# Differences in Characteristics, Hospital Care and Outcomes between Acute Critically Ill Emergency Department Patients with Early and Late Do-Not-Resuscitate Orders

**DOI:** 10.3390/ijerph18031028

**Published:** 2021-01-25

**Authors:** Julia Chia-Yu Chang, Che Yang, Li-Ling Lai, Ying-Ju Chen, Hsien-Hao Huang, Ju-Sing Fan, Teh-Fu Hsu, David Hung-Tsang Yen

**Affiliations:** 1Department of Emergency Department, Taipei Veterans General Hospital, Taipei 11217, Taiwan; juliahazard@hotmail.com (J.C.-Y.C.); yjchen0304@gmail.com (Y.-J.C.); hhhuang@vghtpe.gov.tw (H.-H.H.); jsfan@vghtpe.gov.tw (J.-S.F.); tfhsu@vghtpe.gov.tw (T.-F.H.); 2School of Medicine, National Yang-Ming University, Taipei 11221, Taiwan; 3Department of Nursing, Taipei Veterans General Hospital, Taipei 11217, Taiwan; cyang@vghtpe.gov.tw (C.Y.); lllai@vghtpe.gov.tw (L.-L.L.); 4Institute of Emergency and Critical Care Medicine, College of Medicine, National Yang-Ming University, Taipei 11221, Taiwan; 5Department of Emergency Medicine, National Defense Medical Center, Taipei 11217, Taiwan

**Keywords:** emergency department, intensive care unit, do-not-resuscitate, hospital care

## Abstract

*Background*: A do-not-resuscitate (DNR) order is associated with an increased risk of death among emergency department (ED) patients. Little is known about patient characteristics, hospital care, and outcomes associated with the timing of the DNR order. *Aim*: Determine patient characteristics, hospital care, survival, and resource utilization between patients with early DNR (EDNR: signed within 24 h of ED presentation) and late DNR orders. *Design*: Retrospective observational study. *Setting/Participants*: We enrolled consecutive, acute, critically ill patients admitted to the emergency intensive care unit (EICU) at Taipei Veterans General Hospital from 1 February 2018, to 31 January 2020. *Results*: Of the 1064 patients admitted to the EICU, 619 (58.2%) had EDNR and 445 (41.8%) LDNR. EDNR predictors were age >85 years (adjusted odd ratios (AOR) 1.700, 1.027–2.814), living in long-term care facilities (AOR 1.880, 1.066–3.319), having advanced cardiovascular diseases (AOR 2.128, 1.039–4.358), “medical staff would not be surprised if the patient died within 12 months” (AOR 1.725, 1.193–2.496), and patients’ family requesting palliative care (AOR 2.420, 1.187–4.935). EDNR patients underwent lesser endotracheal tube (ET) intubation (15.6% vs. 39.9%, *p* < 0.001) and had reduced epinephrine injection (19.9% vs. 30.3%, *p* = 0.009), ventilator support (16.7% vs. 37.9%, *p* < 0.001), and narcotic use (51.1% vs. 62.6%, *p* = 0.012). EDNR patients had significantly lower 7-day (*p* < 0.001), 30-day (*p* < 0.001), and 90-day (*p* = 0.023) survival. *Conclusions*: EDNR patients underwent decreased ET intubation and had reduced epinephrine injection, ventilator support, and narcotic use during EOL as well as decreased length of hospital stay, hospital expenditure, and survival compared to LDNR patients.

## 1. Introduction

Although the main responsibilities of emergency physicians (EPs) when treating acute critically ill emergency department (ED) patients include initial resuscitation, stabilization, rapid diagnosis, and curative treatment, aggressive resuscitation may not be appropriate or desired when managing seriously ill patients with advanced chronic illness with trajectories of dying. Early conversations with patients and surrogates regarding aggressive resuscitative measures are critical with respect to patient autonomy and appropriately tailored care. Do-not-resuscitate (DNR) orders are an alternative for patients at the end of life [1], to prevent nonbeneficial resuscitative measures and unnecessary suffering when patients are imminently dying [2,3]. Placement of DNR orders is variable between different types of hospitals and based on differing patient demographic factors [4,5].

In theory, the DNR order itself should not directly impact care until the moment of cardiac arrest. However, early DNR (EDNR: order placement within 24 h of ED presentation) was found to directly influence both resuscitative and ancillary care [6] and resulted in a decrease in potentially critical hospital interventions, with wide variability in practice patterns between hospitals [3]. EDNR often is a proxy of patient’s underlying disease, prehospital frailty, and burden of comorbidities [4], an independent predictor of 28-day mortality [7]; and a strong predictor of short-term mortality risk [8]. The aim of the study was to determine differences in patient characteristics, demographics, hospital care, survival, and resource utilization between EDNR and late DNR (LDNR: after 24 h of ED admission) among ED patients admitted to the ICU.

## 2. Methods

### 2.1. Study Design

This was a retrospective cohort study of adult ED patients (≥18 years) who presented to the ED of Taipei Veterans General Hospital (TVGH) from 1 February 2018, to 31 January 2020. This project was approved by the TVGH Institutional Research Board, which waived the need for patient consent (IRB No: 2020-11-010BC).

### 2.2. Study Setting

TVGH, a 3000-bed university-affiliated medical center, has an annual ED census of 85,200 ± 1812 over the past five years. The emergency intensive care unit (EICU) is a 13-bed ICU within the ED [9] where acute critically ill patients who are not admitted to the specialized ICU immediately after initial ED resuscitation and stabilization receive intensive care. The primary goal of the EICU setting was to implement continuous emergency and critical quality of care within ED prior to available specialty ICU transfer. The operative system in our EICU is semi-open model, that both the EPs and physicians in other subspecialties cooperatively take care of all admission patients. This system was supervised by Emergency Quality Control Committee.

### 2.3. Patient Population

Patients with a diagnosis meeting the criteria for acute severe critical illness (item A) and who also fulfilled two of the criteria for initiating PCC (item B) were categorized as the PC-eligible group (Table 1). The others were categorized as the PC-ineligible group. Palliative care consultation (PCC) screening was initiated for acute critically ill patients aged ≥18 years who were admitted to the EICU from 1 February 2018, to 31 January 2020. Exclusion criteria were age <18 years and medical records with incomplete or missing data. DNR orders are orders to withhold resuscitative measures including CPR, intubation, defibrillation, cardioactive drugs, or assisted ventilation. Patients who did not sign the DNR form on admission (*n* = 1565) and patients who signed the DNR form before admission (*n* = 185) were excluded (Appendix A). Patients were considered to have a preexisting DNR order if a DNR order was found in the patient’s chart dated before the day of admission. The primary exposure variable was whether an order to limit resuscitation efforts was written within the first 24 h of admission (EDNR).

### 2.4. Palliative Care Assessment and Data Collection

Utilization criteria were formulated by palliative care (PC) and hospice specialists and adopted to identify patients at high risk of poor clinical outcomes as their care commonly involves prolonged use of advanced medical resources or technologies [10]. Two trained authors entered the abstracted data for study analyses. The information, time and date of each DNR orders were collected via inpatient electronic medical record systems.

### 2.5. Outcome Measures

Data collected were patient characteristics, hospital care, medical resource utilization, hospital length of stay (LOS), and total expenditures and in-hospital mortality. Hospital care included endotracheal (ET) intubation and ventilator support, cardiopulmonary resuscitation (CPR), cardioversion/defibrillation, epinephrine injection, vasopressor therapy, cardiac pacemaker insertion, extracorporeal membrane oxygenation (ECMO), endotracheal removal, and narcotic use.

### 2.6. Data Analysis

Data are expressed as mean ± SD for continuous variables and number (%) for categorical variables. Data distribution was assessed using the Kolmogorov-Smirnov test. Comparisons of numerical variables were performed using an unpaired *t*-test (parametric data) or Mann-Whitney U test (nonparametric data). Categorical variables were compared using the two-sided chi-square or Fisher’s exact test. Factors showing statistical significance (*p* < 0.05) in the univariate analysis were included in the multiple regression analysis. Survival time was calculated from the date of admission to the date of death using the Kaplan-Meier method, and the difference in survival time between the eligible and ineligible groups was compared using the log-rank test. *p* < 0.05 was considered statistically significant. Statistical analysis was performed using SPSS software version 22.0 (SPSS Inc., Chicago, IL, USA).

## 3. Results

A total of 1064 patients were recruited for the study; 619 (58.2%) had EDNR and 445 (41.8%) LDNR. The screening items for PC consultation at the time of EICU admission are shown in Table 1. Patients with EDNR had more advanced cardiovascular diseases, advanced central neurological diseases, septic shock, adult respiratory distress syndrome (ARDS), multiple organ failure or impending death, and were very severely frail (all *p* < 0.001). The clinical characteristics of EDNR and LDNR patients are compared in Table 2. The mean age of EDNR patients were older than LDNR (80.8 vs. 77.3 years, *p* < 0.001). While more LDNR patients lived with family (84.9% vs. 78.2%), more EDNR patients lived in veterans’ homes (4.4% vs. 2.3%) and long-term care facilities (8.6% vs. 4.7%, all *p* = 0.035). Patients with EDNR had reduced length of hospital stay (17.8 ± 18.4 days vs. 30.3 ± 31.7 days, *p* < 0.001), and lower total hospital expenses (246,684 ± 266,447 new Taiwan dollar (NTD) vs. 468,532 ± 476,382, *p* < 0.001).

Table 3 shows the univariate and multiple logistic regression analyses of clinical characteristics between EDNR and LDNR patients. The risk factors associated with EDNR were age >85 years (AOR 1.700, *p* = 0.039), living in long-term care facilities (AOR 1.880, *p* = 0.029), presence of advanced cardiovascular diseases (A5) (AOR 2.128, *p* = 0.039), patients whom medical staff would not be surprised if they died within 12 months (B1) (AOR 1.725, *p* = 0.004), and patients whose family requested PC (B8) (AOR 2.420, *p* = 0.015). The screening items (item A and item B) for assessment of palliative care consultation at the time of admission were listed in full in Table 1.

Differences in hospital care between EDNR and LDNR patients are compared in Table 4. Patients with EDNR received less endotracheal tube (ET) intubation procedures (15.6% vs. 39.9%, *p* < 0.001), less epinephrine injection (19.9% vs. 30.3%, *p* = 0.009), less ventilator support (16.7% vs. 37.9%, *p* < 0.001), and less narcotic use (51.1% vs. 62.6%, *p* = 0.012).

Table 5 shows multiple logistic regression analyses of hospital care between patients with mortality with EDNR and LDNR. Patients with EDNR underwent lesser ET intubation procedures (AOR 0.198, *p* = 0.007) and had reduced narcotic use (AOR 0.518, *p* = 0.001).

Figure 1 shows the Kaplan-Meier curve for survival between patients with EDNR and LDNR. Patients with EDNR had a significantly lower 7-day, 30-day and 90-day survival.

## 4. Discussion

The study found several differences in patient characteristics, hospital care, survival, and resource utilization between EDNR and LDNR patients.

### 4.1. Characteristics

Patient characteristics that predict EDNR were age >85 years, living in long-term care facilities, presence of advanced cardiovascular diseases (A5), “medical staff would not be surprised if the patient died within 12 months of this episode” (B1), and patients’ family requesting PC (B8).

### 4.2. Age

Compatible with our finding that age >85 years was an independent risk factor for EDNR, other studies also found that older patients were more likely to have an EDNR order [11,12,13]. Age was a powerful predictor of an explicit DNR directive in all categories of patients older than 50 years of age [13]. Age ≥ 80 years was an independent risk factor for DNR orders after controlling for comorbid conditions [14]. Other than being associated with more comorbidity, functional impairment, and higher mortality [14], older patients may have an opportunity to discuss with their physicians and families about advance directives and may be more likely to have accepted and expected their own mortality [15]. However, if decisions on EDNR are based purely on the patient’s chronological age without factoring in survival, quality of life, or patients’ wishes, it may be constituted as ageism. Our study confirmed that age is an important factor for EDNR in critically ill patients, but whether ageism, withholding treatment solely on the basis of age, plays a part in the decision-making process remains unclear.

### 4.3. Living in Long-Term Care Facilities

Our study found that patients living in long-term care facilities were more likely to have EDNR orders (AOR 1.880). Of the 74 patients from nursing homes in our study who signed a DNR order at the ED, 53 patients (71.62%) had EDNR. Similarly, a Danish cohort of patients with community-acquired pneumonia (CAP) found patients with EDNR were older and more frequently nursing home residents (41% vs. 6%, *p* < 0.001). [16] Marrie et al., found that coming from a chronic care facility or a nursing home was a major demographic associated with DNR upon admission, and more than half (53.8%) from institutions had a DNR order in place on admission [17]. This may reflect nursing home policies or a greater awareness among this group to have advanced directives. However, the prevalence of DNR directives among Taiwanese nursing home residents was lower than that in other countries [18]. The EDNR status associated with long-term care facilities may be due to physicians’ awareness of the poor outcomes of resuscitation for nursing home patients and lower odds of achieving return of spontaneous circulation (ROSC) [19] and more likely to approach family and surrogates early with DNR discussion.

### 4.4. Advanced Cardiovascular Disease

Our study found that presence of advanced cardiovascular diseases (chronic heart failure (CHF, New York Heart Association III or IV), chest pain, or dyspnea while performing minimal exercise or on minimal exertion, or devastating inoperable peripheral vascular diseases) were a strong predictor of EDNR (AOR 2.128). EOL discussion in patients with heart failure (HF) is of particular importance because patients often experience repeated hospitalizations and a progressive decline in quality of life as they approach death [20]. However, the waxing and waning pattern typical in HF makes it difficult to accurately prognosticate expected survival, rendering it difficult for physicians to approach patients and surrogates with DNR discussions. Moreover, patients with HF were found to have frequent changes in code status, underscoring the importance of periodically reviewing resuscitation preferences as advocated by the American Heart Association [21]. A study found that three-quarters of community patients with HF elect DNR before death; however, changes in resuscitation preference are often made in the hospital in the final days to weeks of life [22]. This discordance with our finding that patients with HF are at higher risk for EDNR may be because our study included patients who were older (mean 80.8 ± 14.2 years) and had more advanced disease (CCI ≥ 7 47.8%). This may explain the early DNR order in CHF patients with advanced age that tend to underestimate their life expectancy [23].

### 4.5. Medical Staff Would Not Be Surprised If the Patient Died within 12 Months

The decision to forgo resuscitative measures should reflect patient values and preferences. However, physicians’ judgments on patient condition and survival may have a direct impact on patients’ preference for DNR decisions. Many patients and surrogates require a discussion on prognosis with their physicians prior to making a DNR decision [24]. Our study found that the factor “medical staff would not be surprised if the patient died within 12 months of this episode” was a risk factor for EDNR. In line with our study results, a multicenter study found that the one of the strongest predictors of DNR directives were physician prediction of low probability of survival. It is not only physician predictions of high likelihood of death that were associated with DNR order but also moderate likelihood of death [12]. The question “should physician’s judgments on patient survival influence DNR decision?”, is an ethical dilemma.

### 4.6. Patients’ Family Requesting Palliative Care

Our study found that “patients’ family requesting palliative care” is a predictor of EDNR. Inability to participate in decision-making was a strong predictor of a DNR directive during the first 24 h of ICU admission [13]. Patients who were unable to participate in decision-making were significantly more likely to have a DNR directive than a resuscitate directive [13]. A Taiwanese study revealed that the prevalence of DNR directives among Taiwanese nursing home residents was lower than in other countries, with 91% of the directives being put in place by family surrogates [18]. Other studies have shown that as many as 40% of hospitalized adults are unable to make their own medical decisions [25], with DNR decisions being made by family one-third of the time [26]. This is consistent with our finding that 59.4% of patients with DNR and 68.5% with EDNR were categorized as requiring assistance in terms of medical decisions, who were unable to participate in DNR decision making.

### 4.7. Hospital Care

Our study found that EDNR is associated with decreased ET intubation, epinephrine injection, ventilator support, and narcotic use, but no difference in CPR, cardioversion, vasopressor use, cardiac pacemaker insertion, ECMO, intraaortic balloon pumping (IABP), or withdrawal of ET tube was found compared to that in LDNR. This is similar to a sepsis study where the DNR group did not receive less ancillary care of the central line, vasopressors, blood transfusion, emergent hemodialysis, or surgery [11]. Our findings are also comparable with a study where chronic obstruction pulmonary disease (COPD)decedents with EDNR were less likely to undergo invasive mechanical ventilator support during their terminal hospitalization [27]. In theory, the DNR order itself should not directly impact care until the moment of cardiac arrest. However, one study found that EDNR directly influenced both resuscitative and ancillary care, with fewer invasive interventions being performed in the last week of life, including dialysis, mechanical ventilation, feeding tubes, and CPR, compared to those with LDNR and no DNR [6]. Another study on out-of-hospital cardiac arrest (OHCA) patients found that EDNR is associated with a significant decrease in potentially critical therapeutic options, including cardiac catheterization, bypass surgery, and blood transfusion after resuscitation, and is associated with less aggressive hospital care, fewer potentially beneficial procedures, and worse survival [3]. The impact of EDNR on both resuscitative measures and ancillary care may be that patients with DNR, are also less likely to receive nonbeneficial aggressive care at the end of life [28] and are more likely to receive care consistent with their preferences [29]. This wide variability in practice patterns between hospitals and physicians suggests a lack of standardized approach to the EDNR order and subsequent resuscitative measures and ancillary care. This further emphasizes the importance of communication between physicians and patients to align care with treatment goals. Physicians should be careful not to interpret DNR, which is “do not perform CPR in the event of cardiac arrest” as “do not actively treat this patient.”

In contrast to a study in which nurses were more comfortable giving opioids for pain management at the EOL [6], our study found that EDNR is associated with decreased narcotic use. We hypothesize that the shorter duration from DNR placement to death and shorter hospital LOS associated with EDNR allowed less time for physicians to address the family members on patients’ comfort during the care. This certainly leaves room for improvement in patient comfort during hospital care, especially in patients with an EDNR order.

### 4.8. Survival

Patients with EDNR had lower 7-, 30-, and 90-day survival; our finding is compatible with another study where EDNR was found to be an independent predictor for 28-day mortality [7]. In a Danish cohort of patients with community acquired pneumonia (CAP), EDNR was associated with higher mortality after adjustment for clinical risk factors [16]. Among intracerebral hemorrhage (ICH) patients, EDNR is an independent predictor of poor outcome [30,31]; 2.6 times more likely to die than those without DNR order [30]. OHCA patients with EDNR usually die in the hospital without discharge to home within one day of admission [3]. Moreover, EDNR order is often a proxy of the patient’s underlying disease, prehospital frailty, and burden of comorbidities [4] that is not captured by the established risk factors for mortality. Hence, EDNR may function as a composite of prognostic variables, resulting in a strong prediction of short-term mortality risk [8]. However, other studies have argued otherwise. In one study, patients with LDNR (written on day 6 or later) were twice as likely to die in the hospital than patients with EDNR. [32] Marrie et al., hypothesized that EDNR reflects comorbidities and the general health status of a patient, while LDNR represents a lack of response to treatment and comorbidity [17]. The LDNR discussion that occurred later may be because patients are not responding to treatment and at imminent risk of death. A study on sepsis, in fact, found better outcomes in the EDNR group than that in the LDNR group [11]. It remains unclear if the EDNR in our study directly reflects general health status or LDNR directly reflects a lack of response to treatment. The nature and breadth of the discussion leading to EDNR, LDNR, and subsequent withdrawal of care are beyond the scope of the study, leaving room for further research.

### 4.9. Resource Utilization

Patients with EDNR had decreased hospital length of stay (LOS) (17.8 ± 18.4 days vs. 30.3 ± 31.7 days) and decreased total medical expenditure (246,684 ± 266,447 NTD vs. 468,532 ± 476,382) compared to LDNR patients, which is compatible with other studies. One study on COPD patients found that the average total medical cost during the last hospitalization was nearly twofold greater for LDNR than for EDNR decedents [27]. We further stratified patients into mortality and survival groups. Among patients with mortality, the hospital LOS in EDNR patients was shorter than LDNR. (11.7 ± 16.4 days vs. 25.2 ± 24.5 days, *p* < 0.001). The lower probability of survival in EDNR compared to LDNR (Figure 1) may explain the shorter hospital LOS. Surprisingly however, among patients who survived, EDNR had a shorter hospital LOS (faster recovery) than LDNR (22.9 ± 18.5 vs. 3.4 ± 36, *p* < 0.001). The finding agrees with Marrie et al.’s hypothesis that LDNR may be a result of poor response to treatment and comorbidity [17] hence patients with LDNR who survived may have a prolonged hospital course.

## 5. Limitations

Our study has several limitations. First, as a retrospective study, it is liable to underreport due to missing or incomplete medical records. Second, although the inclusion criteria were strictly followed, there still may exist confounding discrepancies between the criteria and clinical conditions of patients recruited. Third, although the study determined the differences in hospital care and outcome between EDNR and LDNR, it did not stratify patients into subgroups based on diagnoses such as ICH, OHCA, CAP, etc. Fourth, the study determined the patient characteristics associated with EDNR, but the rationale for selecting EDNR and LDNR remains unclear in these patients. Fifth, the study did not assess psychosocial aspects such as patient and surrogate viewpoints, their satisfaction or dissatisfaction, and patients’ quality of death associated with EDNR.

## 6. Conclusions

Physicians should understand the potential impact on hospital care and survival associated with EDNR to tailor care with treatment goals. Patients who have not had this discussion should be made aware that EDNR is associated with decreased ET intubation, reduced epinephrine injection, ventilator support, and narcotic use during hospital care, decreased length of hospital stay, hospital expenditure, and decreased survival.

While EDNR orders may be appropriate for guiding subsequent treatment, EDNR may not be entirely appropriate in certain groups of patients in whom long-term prognosis is difficult to ascertain and 24 h may be premature to make this decision.

## Figures and Tables

**Figure 1 ijerph-18-01028-f001:**
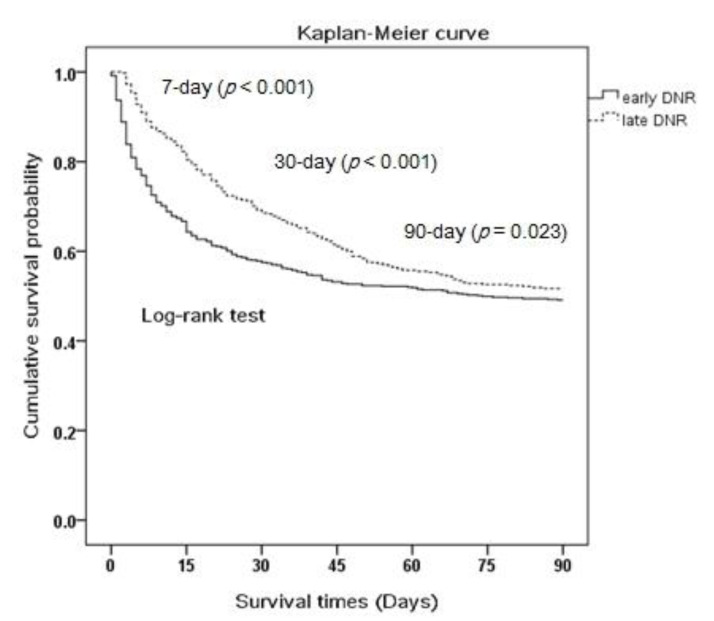
The cumulative survival curve of early DNR patients and late DNR patients.

**Table 1 ijerph-18-01028-t001:** The screening items for needs assessment of palliative care consultation among 1064 patients at the time of admission.

Items	Overall *n* = 1064 (%)	Early DNR *n* = 619 (%)	Late DNR *n* = 445 (%)	*p*-Value
A. Acute critical and life-limiting illness				
1. Advanced cancer, metastatic or locally aggressive disease	204 (19.2)	123 (19.9)	81 (18.2)	0.495
2. Advanced COPD who needs long-term oxygen therapy or respiratory failure requiring assisted ventilation	11 (1.0)	7 (1.1)	4 (0.9)	0.712
3. End-stage liver disease, e.g., cirrhosis, that repeatedly appears with jaundice, ascites, peritonitis, hepatic coma, esophageal varices	20 (1.9)	14 (2.3)	6 (1.3)	0.279
4. Acute or chronic renal failure, decision of not receiving dialysis	26 (2.4)	19 (3.1)	7 (1.6)	0.119
5. Advanced cardiovascular diseases (chronic heart failure NYHA III or IV, chest pain, or dyspnea while in minimal exercise or exertion, or devastating inoperable peripheral vascular diseases) *	60 (5.6)	48 (7.8)	12 (2.7)	<0.001
6. Advanced central neurological diseases (e.g., stroke, dementia) in long-term bed-bound, combined with repeatedly or severely progressive deterioration or recurrent pneumonia, shortness of breath, or respiratory failure requiring hospital admission *	295 (27.7)	202 (32.6)	93 (20.9)	<0.001
7. Septic shock, ARDS, multiple organ failure, or impending death (other devastating diseases) *	343 (32.2)	231 (37.3)	112 (25.2)	<0.001
8. Very severely frail (completely dependent, approaching the end-of-life, CSHA-CFS > scale 8 and 9) *	61 (5.7)	49 (7.9)	12 (2.7)	<0.001
B. The unmet palliative care needs				
1. Medical care staffs would not be surprised if the patient died within 12 months of this episode (surprise question) *	544 (51.1)	374 (60.4)	170 (38.2)	<0.001
2. Appearing progressive functional deterioration with ≥3 ADLs needing for assistance	363 (34.1)	226 (36.5)	137 (30.8)	0.052
3. Appearing biopsychosocial discomforts needing hospital admission *	368 (34.6)	247 (39.9)	121 (27.2)	<0.001
4. Patients with three or more unexpected emergency department visits or hospital admissions within 6 months, with symptoms consistent with a terminal or degenerative chronic medical condition *	235 (22.1)	159 (25.7)	76 (17.1)	0.001
5. Patients Weight loss 10% or BMI ≤ 18 within 6 months	10 (0.9)	8 (1.3)	2 (0.4)	0.160
6. Bed-bound patients with long-term unhealed bed sore or ulceration *	54 (5.1)	39 (6.3)	15 (3.4)	0.032
7. Needing complicated medical care and assistance of medical decisions, including do-not-resuscitate order, ventilator, or nutritional supports *	632 (59.4)	424 (68.5)	208 (46.7)	<0.001
8. Patient’s family request of palliative care *	53 (5.0)	41 (6.6)	12 (2.7)	0.004

Results expressed as number (%) for categorical variables; Of the 1064 screened patients, 36.7% (390/1064) has one item, 27.1% (288/1064) has 2 items, and 1.7% (18/1064) has 3 items of acute critical and life-limiting illnesses. Of the 1064 patients having one or more items of acute critical and life-limiting illnesses, 1.2% (13/1064) has one item, 9.4% (100/1064) has 2 items, 28% (298/1064) has 3 items, 22.9% (244/1064) has 4 items, and; 3.3% (35/1064) has 5 or more items of the unmet palliative care needs; COPD = chronic obstructive pulmonary disease; NYHA = New York Heart Association; ARDS = adult respiratory distress syndrome; CSHA-CFS = Chinese-Canadian study of health and aging clinical frailty scale ADL = activities of daily living; BMI = body mass index; * *p* < 0.05 is considered statistically significant using chi-squared analysis or Fisher’s exact test.

**Table 2 ijerph-18-01028-t002:** Comparison of clinical characteristics between early DNR and late DNR patients.

Variable	Overall	Early DNR	Late DNR	*p*-Value
	*n* = 1064 (%)	*n* = 619 (%)	*n* = 445 (%)	
Age, year *	80.8 ± 14.2	83.3 ± 12.8	77.3 ± 15.4	<0.001
<65	157 (14.8)	73 (11.8)	84 (18.9)	
65–75	137 (12.9)	54 (8.7)	83 (18.7)	
75–85	219 (20.6)	122 (19.7)	97 (21.8)	
>85	551 (51.8)	370 (59.8)	181 (40.7)	
Female sex	399 (37.5)	234 (37.8)	165 (37.1)	0.81
Insurance status *				0.003
National health insurance only	621 (58.4)	338 (54.6)	283 (63.6)	
With Medicaid	443 (41.6)	281 (45.4)	162 (36.4)	
Living conditions *				0.035
With family	860 (81.0)	484 (78.2)	376 (84.9)	
Veterans home	37 (3.5)	27 (4.4)	10 (2.3)	
Long-term care facilities	74 (7.0)	53 (8.6)	21 (4.7)	
Solitary living	70 (6.6)	42 (6.8)	28 (6.3)	
Others	21 (2.0)	13 (2.1)	8 (1.8)	
Marital status				0.287
Single	93 (8.8)	54 (8.7)	39 (8.8)	
Married	642 (60.6)	363 (58.7)	279 (63.3)	
Divorced	37 (3.5)	20 (3.2)	17 (3.9)	
Widow or widower	387 (27.1)	181 (29.3)	106 (24.0)	
Religion				0.478
Taoism	189 (17.8)	108 (17.4)	81 (18.3)	
Buddhism	355 (33.5)	205 (33.1)	150 (33.9)	
Catholic/Christian	97 (9.1)	53 (8.6)	44 (10.0)	
Others	11 (1.0)	9 (1.5)	2 (0.5)	
None	409 (38.5)	244 (39.4)	165 (37.3)	
Educational level				0.055
Higher than high school	444 (42.0)	244 (39.5)	200 (45.5)	
Lower than high school	613 (58.0)	373 (60.5)	240 (54.5)	
Current alcohol consumption	18 (1.7)	11 (1.8)	7 (1.6)	0.796
Current smoker *	69 (6.5)	32 (5.2)	37 (8.3)	0.041
TTAS *				0.004
Emergent (triage 1)	404 (38.1)	258 (41.9)	146 (32.9)	
Urgent (triage 2)	371 (35.0)	212 (34.4)	159 (35.8)	
Non-urgent (triage 3, 4)	285 (26.9)	146 (23.7)	139 (31.3)	
Glasgow Coma Scale *	10.8 ± 4.5	10.4 ± 4.6	11.4 ± 4.2	<0.001
13–15	536 (50.4)	291 (47.0)	245 (55.1)	
5–12	366 (34.4)	214 (34.6)	152 (34.2)	
3–4	162 (15.2)	114 (18.4)	48 (10.8)	
Mean blood pressure in the emergency department (ED) (mmHg)	91.1 ± 24.2	90.6 ± 24.3	91.9 ± 24.0	0.369
Charlson Comorbidity Index	6.7 ± 2.5	6.8 ± 2.3	6.6 ± 2.7	0.414
≤3	59 (5.5)	20 (3.2)	39 (8.8)	
4–6	496 (46.6)	298 (48.1)	198 (44.5)	
≥7	509 (47.8)	301 (48.6)	208 (46.7)	
APACHE II score at admission *	21.8 ± 8.4	22.7 ± 8.4	20.6 ± 8.2	<0.001
0–14	204 (19.2)	101 (16.3)	103 (23.1)	
15–24	486 (45.7)	277 (44.7)	209 (47.0)	
>24	374 (35.2)	241 (38.9)	133 (29.9)	
Hospital length of stay (day) *	23.1 ± 25.6	17.8 ± 18.4	30.3 ± 31.7	<0.001
Total hospital expense (point) *	339,468 ± 384,772	246,684 ± 266,447	468,532 ± 476,382	<0.001
Inhospital mortality	480 (45.1)	282 (45.6)	198 (44.5)	0.731

Results expressed as number (%) for categorical variables and mean ± standard deviation for numerical variables; TTAS = Taiwan Triage and Acuity Scale; ED = emergency department; APACHE = Acute Physiology and Chronic Health Evaluation; ICU = intensive care unit; * *p* < 0.05 is considered statistically significant using Mann-Whitney U test or chi-squared analysis.

**Table 3 ijerph-18-01028-t003:** Univariate and multiple logistic regression analyses of clinical characteristics between early DNR and late DNR patients.

Variable	Univariate Analysis			Multiple Logistic Regression		
	OR	95% CI	*p*	AOR	95% CI	*p*
Age, year						
<65	1			1		
65–75	0.749	(0.470–1.191)	0.222	0.677	(0.399–1.151)	0.15
75–85	1.447	(0.959–2.184)	0.078	1.201	(0.725–1.992)	0.477
>85	2.352	(1.640–3.373)	<0.001	1.7	(1.027–2.814)	0.039 *
Insurance status						
National health insurance only	1			1	1	
With Medicaid	1.452	(1.131–1.864)	0.003	0.988	(0.729–1.340)	0.939
Living conditions						
With family	1			1	1	
Veterans home	2.098	(1.003–4.108)	0.046	1.707	(0.753–3.866)	0.2
Long-term care facilities	1.961	(1.162–3.308)	0.012	1.88	(1.066–3.319)	0.029 *
Solitary living	1.165	(0.709–1.915)	0.546	1.544	(0.896–2.659)	0.118
Others	1.262	(0.518–3.077)	0.608	1.533	(0.591–3.982)	0.38
Smoker						
No smoking/quit smoking	1			1		
Current smoker	0.602	(0.369–0.983)	0.042	0.848	(0.497–1.447)	0.545
TTAS						
Non-urgent	1			1		
Urgent	1.269	(0.931–1.731)	0.132	1.162	(0.830–1.628)	0.382
Emergent	1.682	(1.236–2.290)	0.001	1.413	(1.005–1.985)	0.046
Glasgow Coma Scale						
13–15	1			1		
5–12	1.185	(0.906–1.550)	0.215	0.816	(0.590–1.128)	0.218
3–4	2	(1.371–2.917)	<0.001	1.434	(0.924–2.226)	0.108
Charlson Comorbidity Index						
≤3	1			1		
4–6	2.935	(1.663–5.180)	<0.001	1.53	(0.780–3.000)	0.216
≥7	2.822	(1.600–4.976)	<0.001	1.44	(0.732–2.832)	0.291
APACHE II score at admission						
0–14	1			1		
15–24	1.352	(0.974–1.876)	0.072	1.042	(0.713–1.524)	0.83
>24	1.848	(1.307–2.613)	0.001	1.167	(0.758–1.798)	0.483
palliative care consultation screening items						
A5	3.033	(1.592–5.780)	0.001	2.128	(1.039–4.358)	0.039 *
A6	1.833	(1.381–2.435)	<0.001	0.955	(0.659–1.383)	0.807
A7	1.77	(1.353–2.317)	<0.001	0.956	(0.673–1.357)	0.8
A8	3.102	(1.630–5.903)	0.001	1.674	(0.839–3.342)	0.144
B1	2.469	(1.923–3.171)	<0.001	1.725	(1.193–2.496)	0.004 *
B3	1.778	(1.366–2.314)	<0.001	1.181	(0.844–1.652)	0.331
B4	1.678	(1.236–2.278)	0.001	1.127	(0.791–1.606)	0.508
B6	1.928	(1.049–3.542)	0.035	1.194	(0.615–2.317)	0.601
B7	2.478	(1.926–3.187)	<0.001	1.279	(0.815–2.007)	0.284
B8	2.56	(1.329–4.929)	0.005	2.42	(1.187–4.935)	0.015 *

TTAS = Taiwan Triage and Acuity Scale; APACHE = Acute Physiology and Chronic Health Evaluation; OR = odds ratio; 95% CI = 95% confidence interval; AOR = adjusted odds ratio; * *p* < 0.05 is considered statistical significance in regression model.

**Table 4 ijerph-18-01028-t004:** Comparison of hospital care in hospitalization between 282 early DNR patients with mortality and 198 late DNR patients with mortality.

Variable	Early DNR Patients with Mortality	Late DNR Patients with Mortality	*p*
	*n* = 282 (%)	*n* = 198 (%)	
Place of death			0.446
Intensive care unit	109 (38.7)	73 (36.9)	
Wards	108 (38.3)	89 (44.9)	
Hospice unit	31 (11.0)	17 (8.6)	
Critical against advice discharge	34 (12.1)	19 (9.6)	
End-of-life care			
ET intubation *	44 (15.6)	79 (39.9)	<0.001
CPR	12 (4.3)	16 (8.1)	0.078
Epinephrine *	56 (19.9)	60 (30.3)	0.009
Cardioversion or defibrillation	7 (2.5)	5 (2.5)	0.976
Vasopressors	183 (64.9)	138 (69.7)	0.271
Cardiac pacemaker	3 (1.1)	1 (0.5)	0.507
Ventilator support *	47 (16.7)	75 (37.9)	<0.001
ECMO or IABP	1 (0.4)	4 (2.0)	0.077
Withdrawal of ET tube	11 (3.9)	10 (5.1)	0.544
Narcotics use *	144 (51.1)	124 (62.6)	0.012

Results expressed as number (%) for categorical variables; ET = endotracheal; CPR = cardiopulmonary resuscitation; ECMO = extracorporeal membrane oxygenation; IABP = intra-aortic balloon pump; * *p* < 0.05 is considered statistically significant using chi-squared analysis or Fisher’s exact test.

**Table 5 ijerph-18-01028-t005:** Multiple logistic regression analyses of hospital care between early DNR and late DNR patients with mortality

Variable	Univariate Analysis			Multiple Logistic Regression		
	OR	95% CI	*p*	AOR	95% CI	*p*
ET intubation *	0.278	(0.181–0.428)	<0.001	0.198	(0.061–0.643)	0.007
Epinephrine	0.570	(0.374–0.868)	0.009	0.639	(0.404–1.010)	0.055
Ventilator support	0.328	(0.214–0.502)	<0.001	1.460	(0.449–4.752)	0.529
Narcotics use *	0.623	(0.430–0.902)	0.012	0.518	(0.347–0.772)	0.001

ET = endotracheal; OR = odds ratio; 95% CI = 95% confidence interval; AOR = adjusted odds ratio; * *p* < 0.05 is considered statistical significance in regression model.

## Data Availability

All data submitted comply with Institutional or Ethical Review Board requirements and applicable government regulations. For further information, please contact David Hung-Tsang Yen (hjyen@vghtpe.gov.tw).

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
