# Peer review of "Differences in Characteristics, Hospital Care and Outcomes between Acute Critically Ill Emergency Department Patients with Early and Late Do-Not-Resuscitate Orders"

_ijerph, 2021, doi:10.3390/ijerph18031028_

Round 1

Reviewer 1 Report

Dear authors,

Thank you for your study, it has important results with important practical implications.

My suggestions for correction:

  • references in the body of manuscript should be prepared according to Journal guidance;
  • in tables you used an asterisk, however under the table, in legend it shoud be given again (*) with expalanation;
  • I suggest to include more information regarding the process of data collection (medical records analysis) - who, how, when... It should be done in methodology part, a short and consistent way as manuscript is long.
  • in a discussion - few sentences should be included highlighting the ethical aspects, especially regarding EDNR

Author Response

  1. References in the body of manuscript should be prepared according to Journal guidance.

Answer: We are grateful to the reviewer’s thoughtful comments and edited the references in the body of manuscript.

  1. In tables you used an asterisk, however under the table, in legend it shoud be given again (*) with expalanation

Answer: We are grateful to the reviewer’s thoughtful comments and highlighted the explanation. (p. 4 line 129-130; p.5 line 132-135; p.6 line 145-146; p. 6 line 154-155, , p. 7 line 161-162)

  1. I suggest to include more information regarding the process of data collection (medical records analysis) - who, how, when... It should be done in methodology part, a short and consistent way as manuscript is long.

Answer: We are grateful to the reviewer’s thoughtful comments and added more information and add high-light to clarify the method section. (p.2-3 line 57-96)

2.2. Study setting

TVGH, a 3000-bed university-affiliated medical center, has an annual ED census of 85,200 ± 1,812 over the past five years. The emergency intensive care unit (EICU) is a 13-bed ICU within the ED9 where acute critically ill patients who are not admitted to the specialized ICU immediately after initial ED resuscitation and stabilization receive intensive care. The primary goal of the EICU setting was to implement continuous emergency and critical quality of care within ED prior to available specialty ICU transfer. The operative system in our EICU is semi-open model, that both the EPs and physicians in other subspecialties cooperatively take care of all admission patients. This system was supervised by Emergency Quality Control Committee.

2.3. Patient population

Patients with a diagnosis meeting the criteria for acute severe critical illness (item A) and who also fulfilled two of the criteria for initiating PCC (item B) were categorized as the PC-eligible group (table 1).The others were categorized as the PC-ineligible group. Palliative care consultation (PCC) screening was initiated for acute critically ill patients aged ³18 years who were admitted to the EICU from February 1, 2018, to Jan 31, 2020. Exclusion criteria were age <18 years and medical records with incomplete or missing data. DNR orders are orders to withhold resuscitative measures including CPR, intubation, defibrillation, cardioactive drugs, or assisted ventilation.Patients who did not sign the DNR form on admission (n=1,565) and patients who signed the DNR form before admission (n=185) were excluded (appendix 1). Patients were considered to have a preexisting DNR order if a DNR order was found in the patient’s chart dated before the day of admission. The primary exposure variable was whether an order to limit resuscitation efforts was written within the first 24 h of admission (EDNR).

2.4. Palliative care assessment and data collection

Utilization criteria were formulated by palliative care (PC) and hospice specialists and adopted to identify patients at high risk of poor clinical outcomes as their care commonly involves prolonged use of advanced medical resources or technologies.10 Two trained authors entered the abstracted data for study analyses. The information, time and date of each DNR orders were collected via inpatient electronic medical record systems.

2.5. Outcome measures

Data collected were patient characteristics, hospital care, medical resource utilization, hospital LOS (length of stay), and total expenditures and in-hospital mortality. Hospital care included endotracheal (ET) intubation and ventilator support, cardiopulmonary resuscitation (CPR), cardioversion/defibrillation, epinephrine injection, vasopressor therapy, cardiac pacemaker insertion, extracorporeal membrane oxygenation (ECMO), endotracheal removal, and narcotic use.

  1. In a discussion - few sentences should be included highlighting the ethical aspects, especially regarding EDNR

Answer: We are grateful to the reviewer’s thoughtful comments and modified the content. The few discussion on ethical aspects were dispersed under subtitles i.e. age, EOL care, medical staff would not be surprised if the patient died within 12 months. (p. 8-9 line 183-187)

4.2. Age

Compatible with our finding that age >85 years was an independent risk factor for EDNR, other studies also found that older patients were more likely to have an EDNR order.11,12,13Age was a powerful predictor of an explicit DNR directive in all categories of patients older than 50 years of age.13 Age ≥80 years was an independent risk factor for DNR orders after controlling for comorbid conditions.14 Other than being associated with more comorbidity, functional impairment, and higher mortality14, older patients may have an opportunity to discuss with their physicians and families about advance directives and may be more likely to have accepted and expected their own mortality.15 However, if decisions on EDNR are based purely on the patient’s chronological age without factoring in survival, quality of life, or patients’ wishes, it may be constituted as ageism. Our study confirmed that age is an important factor for EDNR in critically ill patients, but whether ageism, withholding treatment solely on the basis of age, plays a part in the decision-making process remains unclear.

(p. 9 line 198-201)

4.3. Living in long-term care facilities

Our study found that patients living in long-term care facilities were more likely to have EDNR orders (AOR 1.880). Of the 74 patients from nursing homes in our study who signed a DNR order at the ED, 53 patients (71.62%) had EDNR. Similarly, a Danish cohort on patients with community-acquired pneumonia (CAP) found patients with EDNR were older and more frequently nursing home residents (41% vs. 6%, p < 0.001).16 Marrie et al. found that coming from a chronic care facility or a nursing home was a major demographic associated with DNR upon admission, and more than half (53.8%) from institutions had a DNR order in place on admission.17 This may reflect nursing home policies or a greater awareness among this group to have advanced directives. However, the prevalence of DNR directives among Taiwanese nursing home residents was lower than that in other countries.18 The EDNR status associated with long-term care facilities may be due to physicians’ awareness of the poor outcomes of resuscitation for nursing home patients and lower odds of achieving return of spontaneous circulation (ROSC)19 and more likely to approach family and surrogates early with DNR discussion.

(p.9 228-229)

4.5. Medical staff would not be surprised if the patient died within 12 months

The decision to forgo resuscitative measures should reflect patient values and preferences. However, physicians’ judgments on patient condition and survival may have a direct impact on patients’ preference for DNR decisions. Many patients and surrogates require a discussion on prognosis with their physicians prior to making a DNR decision.24 Our study found that the factor “medical staff would not be surprised if the patient died within 12 months of this episode” was a risk factor for EDNR. In line with our study results, a multicenter study found that the one of the strongest predictors of DNR directives were physician prediction of low probability of survival.12 It is not only physician predictions of high likelihood of death that were associated with DNR order but also moderate likelihood of death. 12 The question “should physician’s judgments on patient survival influence DNR decision?”, is an ethical dilemma.

(p. 10 line 261-265)

Another study on out-of-hospital cardiac arrest (OHCA) patients found that EDNR is associated with a significant decrease in potentially critical therapeutic options, including cardiac catheterization, bypass surgery, and blood transfusion after resuscitation, and is associated with less aggressive hospital care, fewer potentially beneficial procedures, and worse survival.3 The impact of EDNR on both resuscitative measures and ancillary care may be that patients with DNR, are also less likely to receive nonbeneficial aggressive care at the end of life28 and are more likely to receive care consistent with their preferences.29 This wide variability in practice patterns between hospitals and physicians suggests a lack of standardized approach to the EDNR order and subsequent resuscitative measures and ancillary care. This further emphasizes the importance of communication between physicians and patients to align care with treatment goals. Physicians should be careful not to interpret DNR, which is “do not perform CPR in the event of cardiac arrest” as “do not actively treat this patient.”

In contrast to a study in which nurses were more comfortable giving opioids for pain management at the EOL,6 our study found that EDNR is associated with decreased narcotic use. We hypothesize that the shorter duration from DNR placement to death and shorter hospital LOS associated with EDNR allowed less time for physicians to address the family members on patients’ comfort during the care. This certainly leaves room for improvement in patient comfort during hospital care, especially in patients with an EDNR order.

Reviewer 2 Report

The limitations in the construction of the work indicated in the specific paragraph are so relevant that they do not allow the data to be indicated as having scientific significance. In particular, the variability in the collection of data gives the results a simply descriptive character from which no conclusions can be drawn with a background that goes beyond the anecdotal.

Author Response

Reviewer 2

  1. The limitations in the construction of the work indicated in the specific paragraph are so relevant that they do not allow the data to be indicated as having scientific significance. In particular, the variability in the collection of data gives the results a simply descriptive character from which no conclusions can be drawn with a background that goes beyond the anecdotal.

Answer: We are grateful to and agree with the reviewer’s thoughtful comments. The limitations related to the construction of the study render it a descriptive and anecdotal character. Future studies with different study design approach is required to correct this flaw.

Reviewer 3 Report

It is my pleasure to review your paper.

Clearly, you have conceived and completed this study with care and rigour. The report also displays the same qualities in describing method and findings.

However, I believe your research focus is situated within a fairly niche area of practice. This makes it difficult for outsiders to fully appreciate your study aims and conclusions drawn from the data. This may be compounded by lack of information around context to orient the reader, possibly limited by wordcount, and the fact that the manuscript is not in your first language.

I have suggested some areas that could be amended to strengthen the manuscript. Hope you will find them useful.

Page 1: "Hence, palliative care is important... settings." seems at odds with sentences that come before and after. Nothing in your study relates to palliative care.

Page 2: EICU needs elaboration. This is not something commonly encountered outside Taiwan. There are numerous acronyms introduced without the original word spelt in full (at least for the first time). PCC screening is one example. More to follow. Under the section on ‘outcome measures’ – I would disagree that in-hospital mortality is your primary outcome measure here. As for EOL care as another outcome, what you have described are actually not typically referred to as ‘EOL care’. They are more like resuscitation measures or life-sustaining treatments.

Page 3: Which are ‘eligible’ or ‘ineligible’ groups? You described lower length of stay among those with EDNR. Although this is somewhat expected, you are probably referring to both those who were discharged alive and those who eventually died in hospital (yet not specifically stated). Without stratifying these two groups and analysing them separately, I am curious how this apparently frailer group recovered faster than the other group (LDNR).

Page 8: ROSC? Section 4.5 last sentence - unclear what you meant.

Page 9: Section 4.6 last sentence - Same. Unclear. Section 4.8 – CAP & ICH?

Page 10: Section 5 “Second, although the inclusion criteria …” Not sure what you mean.

In summary, this paper is based on a premise that is likely unique to Taiwan, that DNR could be a proxy for minimising inappropriate treatment in a patient close to the end of life. Not only does this constraint interest in this paper, it limits broader generalisability of your study findings. You alluded to this in the last part of section 4.8, where the “nature and breath of discussion leading to EDNR & LDNR” were not apparent in this retrospective review. Although you have cited this as a limitation (page 10, section 5), it remains a major weakness that can only be mitigated using an alternative study design.

That said, the highlight of your paper lies within page 9, section 4.7: “This wide variability in practice patterns … do not actively treat this patient.” It is clearly demonstrated by your study findings and is a key takeaway for any reader.

Thank you.

Author Response

  1. Page 1: "Hence, palliative care is important... settings." seems at odds with sentences that come before and after. Nothing in your study relates to palliative care.

Answer: We are grateful to the reviewer’s thoughtful comments and deleted the sentence as it seems quite out of place. (p.1 line 42-43)

Early conversations with patients and surrogates regarding aggressive resuscitative measures are critical with respect to patient autonomy and appropriately tailored care. Hence, palliative care (PC) is especially important in the ED and intensive care unit (ICU) settings.

Page 2: EICU needs elaboration. This is not something commonly encountered outside Taiwan.

Answer: We are grateful to the reviewer’s thoughtful comments and modified the content acc2.2. Study setting. (p.2 line 64-71)

TVGH, a 3000-bed university-affiliated medical center, has an annual ED census of 85,200 ± 1,812 over the past five years. The emergency intensive care unit (EICU) is a 13-bed ICU within the ED9 where acute critically ill patients who are not admitted to the specialized ICU immediately after initial ED resuscitation and stabilization receive intensive care. The primary goal of the EICU setting was to implement continuous emergency and critical quality of care within ED prior to available specialty ICU transfer. The operative system in our EICU is semi-open model, that both the EPs and physicians in other subspecialties cooperatively take care of all admission patients. This system was supervised by Emergency Quality Control Committee.

In addition In one study, we investigated an intensive care model for acute critically cardiovascular emergency patients in the EICU as compared with those in the coronary care unit (CCU) after ED visits. We demonstrated no statistically significant differences of cumulative survival rates in both the 7-day and in-hospital survival between both groups. (Fu KH, Chen YR, Fan JS, Chen YC, Huang HH, How CK, et al. Emergency department critical care unit for critically ill cardiovascular patients: An observation study. J Chin Med Assoc. 2017;80:233-244.)

There are numerous acronyms introduced without the original word spelt in full (at least for the first time). PCC screening is one example. More to follow.

Answer: We are grateful to the reviewer’s thoughtful comments and modified the content accordingly. Palliative care consultation (PCC); palliative care (PC); endotracheal (ET) LOS (length of stay);new Taiwan dollar (NTD); return of spontaneous circulation (ROSC); heart failure (HF) p.9 intraaortic balloon pumping (IABP); chronic obstruction pulmonary disease (COPD); intracerebral hemorrhage (ICH)

Under the section on ‘outcome measures’ – I would disagree that in-hospital mortality is your primary outcome measure here.

Answer: We are grateful to the reviewer’s thoughtful comments and edited the content accordingly. (p.3 line 91-96)

2.5. Outcome measures

Data collected were patient characteristics, hospital care, medical resource utilization, hospital LOS (length of stay), and total expenditures and in-hospital mortality. Hospital care included endotracheal (ET) intubation and ventilator support, cardiopulmonary resuscitation (CPR), cardioversion/defibrillation, epinephrine injection, vasopressor therapy, cardiac pacemaker insertion, extracorporeal membrane oxygenation (ECMO), endotracheal removal, and narcotic use.

As for EOL care as another outcome, what you have described are actually not typically referred to as ‘EOL care’. They are more like resuscitation measures or life-sustaining treatments.

Answer: We are grateful to the reviewer’s thoughtful comments and modified the content accordingly. What we referred to as “ EOL care” were edited to “hospital care.” in title page, manuscript and tables.

  1. Page 3: Which are ‘eligible’ or ‘ineligible’ groups?

Answer: We are grateful to the reviewer’s thoughtful comments and edited the content accordingly. (p.2 line 72-84)

2.3. Patient population

Patients with a diagnosis meeting the criteria for acute severe critical illness (item A) and who also fulfilled two of the criteria for initiating PCC (item B) were categorized as the PC-eligible group (table 1).The others were categorized as the PC-ineligible group. Palliative care consultation (PCC) screening was initiated for acute critically ill patients aged ³18 years who were admitted to the EICU from February 1, 2018, to Jan 31, 2020. Exclusion criteria were age <18 years and medical records with incomplete or missing data. DNR orders are orders to withhold resuscitative measures including CPR, intubation, defibrillation, cardioactive drugs, or assisted ventilation.Patients who did not sign the DNR form on admission (n=1,565) and patients who signed the DNR form before admission (n=185) were excluded (appendix 1). Patients were considered to have a preexisting DNR order if a DNR order was found in the patient’s chart dated before the day of admission. The primary exposure variable was whether an order to limit resuscitation efforts was written within the first 24 h of admission (EDNR).

You described lower length of stay among those with EDNR. Although this is somewhat expected, you are probably referring to both those who were discharged alive and those who eventually died in hospital (yet not specifically stated). Without stratifying these two groups and analyzing them separately, I am curious how this apparently frailer group recovered faster than the other group (LDNR).

Answer: We are grateful to the reviewer’s thoughtful comments and stratified EDNR and LDNR to compare hospital LOS. (p. 11 line 297-304)

We further stratified patients into mortality and survival groups. Among patients with mortality, the hospital LOS in EDNR patients was shorter than LDNR. (11.7±16.4d vs. 25.2±24.5d, p<0.001). The lower probability of survival in EDNR compared to LDNR (figure 1) may explain the shorter hospital LOS. Surprisingly however, among patients who survived, EDNR had a shorter hospital LOS (faster recovery) than LDNR (22.9± 18.5 vs. 3.4± 36, p<0.001). The finding agrees with Marrie et al.’s hypothesis that LDNR may be a result of poor response to treatment and comorbidity17 hence patients with LDNR who survived may have a prolonged hospital course

Page 8: ROSC? Section 4.5 last sentence - unclear what you meant.

Answer: We are grateful to the reviewer’s thoughtful comments and modified the content accordingly. (p.9 line 198-201)

4.3. Living in long-term care facilities

Our study found that patients living in long-term care facilities were more likely to have EDNR orders (AOR 1.880). Of the 74 patients from nursing homes in our study who signed a DNR order at the ED, 53 patients (71.62%) had EDNR. Similarly, a Danish cohort on patients with community-acquired pneumonia (CAP) found patients with EDNR were older and more frequently nursing home residents (41% vs. 6%, p < 0.001).16 Marrie et al. found that coming from a chronic care facility or a nursing home was a major demographic associated with DNR upon admission, and more than half (53.8%) from institutions had a DNR order in place on admission.17 This may reflect nursing home policies or a greater awareness among this group to have advanced directives. However, the prevalence of DNR directives among Taiwanese nursing home residents was lower than that in other countries.18 The EDNR status associated with long-term care facilities may be due to physicians’ awareness of the poor outcomes of resuscitation for nursing home patients and lower odds of achieving return of spontaneous circulation (ROSC)19 and more likely to approach family and surrogates early with DNR discussion.

Page 9: Section 4.6 last sentence - Same. Unclear.

Answer: We are grateful to the reviewer’s thoughtful comments and modified the content accordingly. (p.10 line 238-241)

4.6. Patients’ family requesting palliative care

Our study found that “patients’ family requesting palliative care” is a predictor of EDNR. Inability to participate in decision-making was a strong predictor of a DNR directive during the first 24 h of ICU admission.13 Patients who were unable to participate in decision-making were significantly more likely to have a DNR directive than a resuscitate directive.13 A Taiwanese study revealed that the prevalence of DNR directives among Taiwanese nursing home residents was lower than in other countries, with 91% of the directives being put in place by family surrogates.18 Other studies have shown that as many as 40% of hospitalized adults are unable to make their own medical decisions,25 with DNR decisions being made by family one-third of the time.26 This is consistent with our finding that 59.4% of patients with DNR and 68.5% with EDNR were categorized as requiring assistance in terms of medical decisions, who were unable to participate in DNR decision making.

Section 4.8 – CAP & ICH?

Answer: We are grateful to the reviewer’s thoughtful comments and modified the content accordingly. (p.10 line 275- 277)

4.8. Survival

Patients with EDNR had lower 7-, 30-, and 90-day survival; our finding is compatible with another study where EDNR was found to be an independent predictor for 28-day mortality.7 In a Danish cohort of patients with community acquired pneumonia (CAP), EDNR was associated with higher mortality after adjustment for clinical risk factors.16 Among intracerebral hemorrhage (ICH) patients, EDNR is an independent predictor of poor outcome;30,31 2.6 times more likely to die than those without DNR order.30

  1. Page 10: Section 5 “Second, although the inclusion criteria …” Not sure what you mean.

Answer: We are grateful to the reviewer’s thoughtful comments and modified the content accordingly. (p.11 line 307-309)

Answer: We are grateful to the reviewer’s thoughtful comments and modified the content accordingly. Our study has several limitations. First, as a retrospective study, it is liable to underreport due to missing or incomplete medical records. Second, although the inclusion criteria were strictly followed, there still may exist confounding discrepancies between the criteria and clinical conditions of patients recruited.

Reviewer 4 Report

Dear authors,

thank you for this interesting manuscript.

Could you clarify the following?

  • introduction
    • what type of DNR do you use, only limiting resuscitation efforts or also other treatment restriction such as a code DNR1-2-3?
  • methods
    • what are the admission criteria for EICU? Are they differnet from the other ICU departments? So coul there be bias in your results? Are patients with higher risk of death not admitted ot the other ICU's?
    • why did you not include the DNR orders prior to the ED? did you rewrite them or reevaluate them? or did these patients were not included at all? if so, why?
  • results
    • Could we have the tables on one page?
    • why did patients with DNR receive ECMO?
    • How many of the patients came from Nursing homes? how many had a DNR prior to admission?
  • discussion
    • is there a reason why you do not mention cancer or renal diease as you go into cardiovascular disease and COPD? rationale?
    • what types of patients had late DNR orders?

please explain all abbreviations: ICH

Author Response

Reviewer 4

  1. Introduction: what type of DNR do you use, only limiting resuscitation efforts or also other treatment restriction such as a code DNR1-2-3?

Answer: We are grateful to the reviewer’s thoughtful comments. DNR orders are orders to withhold resuscitative measures including CPR, intubation, defibrillation, cardioactive drugs, or assisted ventilation. (p.2 line 78-79)

DNR orders are orders to withhold resuscitative measures including CPR, intubation, defibrillation, cardioactive drugs, or assisted ventilation.

  1. Methods: what are the admission criteria for EICU? Are they differnet from the other ICU departments?

Answer: We are grateful to the reviewer’s thoughtful comments and modified the content accordingly. The criteria for EIUC were critically ill patients who are not admitted to specialized ICU immeidately after intial ED resuscitation and stablization. (p.2 line 63-71)

2.2. Study setting

TVGH, a 3000-bed university-affiliated medical center, has an annual ED census of 85,200 ± 1,812 over the past five years. The emergency intensive care unit (EICU) is a 13-bed ICU within the ED9 where acute critically ill patients who are not admitted to the specialized ICU immediately after initial ED resuscitation and stabilization receive intensive care. The primary goal of the EICU setting was to implement continuous emergency and critical quality of care within ED prior to available specialty ICU transfer. The operative system in our EICU is semi-open model, that both the EPs and physicians in other subspecialties cooperatively take care of all admission patients. This system was supervised by Emergency Quality Control Committee.

So could there be bias in your results?

We added the mention of bias in limitation. (p.11 line 315-317)

Sixth, the difference in recruitment criteria between EICU and other subspecialty ICUs is beyond the scope and design of the study. Recruitments of patients admitted EICU served as a selection bias to the study.

Are patients with higher risk of death not admitted to the other ICU's?

In one study, we investigated an intensive care model for acute critically cardiovascular emergency patients in the EICU as compared with those in the coronary care unit (CCU) after ED visits. We demonstrated no statistically significant differences of cumulative survival rates in both the 7-day and in-hospital survival between both groups. (Fu KH, Chen YR, Fan JS, Chen YC, Huang HH, How CK, et al. Emergency department critical care unit for critically ill cardiovascular patients: An observation study. J Chin Med Assoc. 2017;80:233-244.)

  1. Why did you not include the DNR orders prior to the ED? Did you rewrite them or reevaluate them? Or did these patients were not included at all? If so, why?

Answer: We are grateful to the reviewer’s thoughtful comments. The purpose of this paper was to determine differences in care and outcome for acute critically ill patients with an early DNR (first 24 h) vs. late DNR. We restricted this set to patients to those with no prior DNR orders, since the primary exposure variable was whether an order to limit resuscitation efforts was written within the first 24 h of admission (EDNR). Hence we did not include patients who had prior DNR or who rewrote DNR orders after admission.

  1. Results: Could we have the tables on one page?

Answer: We are grateful to the reviewer’s thoughtful comments. We found it difficult to compress the tables into one page, but were diligent in complying to the table restrictions of the manuscript.

  1. Why did patients with DNR receive ECMO?

Answer: We are grateful to the reviewer’s thoughtful comments. Among patients with ECMO, the DNR orders were all signed after an aggressive resuscitative measures followed by poor treatment response. Four out of five patients on ECMO had LDNR and prolong hospital course, compatible with Marrie et al.s hypothesis that LDNR represents a lack of response to treatment and comorbidity.17

  1. How many of the patients came from Nursing homes? How many had a DNR prior to admission?

Answer: We are grateful to the reviewer’s thoughtful comments. Table 2 showed, 74/1043 (7.0%) came from nursing home while 53 (8.6%) had EDNR and 21(4.7%) had LDNR. The primary exposure variable was whether an order to limit resuscitation efforts was written within the first 24 h of admission (EDNR). Hence we did not include patients who had prior DNR or who rewrote DNR orders after admission.

  1. Discussion: is there a reason why you do not mention cancer or renal disease as you go into cardiovascular disease and COPD? rationale?

Answer: We are grateful to the reviewer’s thoughtful comments. We specifically discussed HF, because we found patients with HF had AOR 2.128 of having EDNR (p=0.039). The studies references we cited for discussion on early and late DNR included patients groups of HF, COPD, CAP, OHCA and sepsis.

  1. What types of patients had late DNR orders?

Answer: We are grateful to the reviewer’s thoughtful comments. Table 2 showed patients with LDNR were younger (77.3± 15.4 y/o vs. 83.3± 12.8 y/o p<0.001), 84.9% living with family, 4.7% living in nursing home, had better GCS (11.4 ± 4.2vs. 10.4± 4.6, p<0.001), more with CCI<3 (8.8% vs. 3.2%, p<0.001), fewer with APACHE II score >24 at admission (29.9% vs. 38.9%, p<0.001).

(p. 11, line 281-290) While EDNR may function as a composite of prognostic variables, resulting in a strong prediction of short-term mortality risk.8 Oher studies have argued otherwise. In one study, patients with LDNR (written on day 6 or later) were twice as likely to die in the hospital than patients with EDNR.32 Marrie et al. hypothesized that EDNR reflects comorbidities and the general health status of a patient, while LDNR represents a lack of response to treatment and comorbidity.17 The LDNR discussion that occurred later may be because patients are not responding to treatment and at imminent risk of death. A study on sepsis, in fact, found better outcomes in the EDNR group than that in the LDNR group.11 It remains unclear if the EDNR in our study directly reflects general health status or LDNR directly reflects a lack of response to treatment.

(p.11 line 300-304) Among patients who survived, EDNR had a shorter hospital LOS (faster recover) than LDNR (22.9± 18.5 vs. 3.4± 36, p<0.001). The finding agrees with Marrie et al.’s hypothesis that LDNR may be a result of poor response to treatment and comorbidity hence those LDNR who survived may have a prolonged hospital course.

  1. Please explain all abbreviations: ICH

Answer: We are grateful to the reviewer’s thoughtful comments and modified the content accordingly. Palliative care consultation (PCC); palliative care (PC); endotracheal (ET) LOS (length of stay);new Taiwan dollar (NTD); return of spontaneous circulation (ROSC); heart failure (HF) p.9 intraaortic balloon pumping (IABP); chronic obstruction pulmonary disease (COPD); intracerebral hemorrhage (ICH))

Round 2

Reviewer 2 Report

The limitations already indicated in the previous revision have not been overcome in the new version of the manuscript. The new version does not change the judgment regarding the impossibility of conferring robust scientific significance to the exposed results. This does not allow the manuscript to be considered suitable for publication even in its current revised form. My judgment is that it is rejected.

Author Response

Reviewer 2:

  1. The new version does not change the judgment regarding the impossibility of conferring robust scientific significance to the exposed results. This does not allow the manuscript to be considered suitable for publication even in its current revised form. My judgment is that it is rejected.

Answer: We are grateful to the reviewer’s thoughtful comments and the time spent reviewing and replying to our study. It is a pity the study design does not produce result that qualified for acceptance. Thank you once again.

Reviewer 3 Report

Thank you for sharing your responses and rework of the manuscript. This iteration is a much improved one. On that note, I would agree that it could be published with minor English edicts. However, presentation of the statistical analysis (particularly in tables 3 & 5 on regression analysis) as I said before needed to be clearer. Thanks.

Author Response

  1. I would agree that it could be published with minor English edicts.

Answer: We are grateful to the reviewer’s thoughtful comments. The article was edited for language with a certificate of edit. We will attach the certificate of editing when the manuscript is resubmitted. However as we are allowed only 1 file for upload, we will upload the manuscript for you first. If more editing is required to better clarify the study content, we are ready to comply.

  1. However, presentation of the statistical analysis (particularly in tables 3 & 5 on regression analysis) as I said before needed to be clearer. Thanks.

Answer: We are grateful to the reviewer’s thoughtful comments. For table 3, we added the title “palliative care consultation screening items” within table 3 to make it clearer. We also added to the manuscript in result (p. 5 line 141-142) “The screening items (item A and item B) for assessment of palliative care consultation at the time of admission were listed in full in table 1.” to explain table 3 better.

For Table 5, we modified the title to “Table 5. Multiple logistic regression analyses of hospital care between early DNR and late DNR patients with mortality” to better clarify the regression analysis. We also added to the manuscript (p.9 line 159-160) “Table 5 shows multiple logistic regression analyses of hospital care between patients with mortality with EDNR and LDNR.
